# Topological Alignment: A Universal Framework for Anomaly Detection

## Abstract

Zero-Shot Anomaly Detection (ZS-AD) methods based on Vision-Language Models face a critical vulnerability: a paradoxical performance collapse when trained on large-scale, diverse data. We identify this phenomenon as Negative Transfer in Domain Generalization (NTDG) and diagnose its root cause as a Domain Conflict: a fundamental structural incompatibility where a single, rigid geometric decision boundary fails to separate topologically complex data manifolds from multiple domains. To escape this trap, we propose a paradigm shift from geometric separation to robust topological separability, actualized in our TDA-CLIP framework. The framework introduces two general-purpose, plug-and-play topological tools: (1) a macro-level Homology Consistency Loss ($\mathcal{L}_{\mathrm{HC}}$) that acts as a structural regularizer to enforce a globally consistent feature space, and (2) a micro-level Topology-Guided Attention (TGA) module that purifies features by amplifying salient local evidence. Crucially, these topological components are active only during training and are completely pruned at inference time, delivering substantial performance gains while introducing absolutely no extra cost at inference. Extensive experiments demonstrate that our framework is the first to overcome this negative transfer, consistently benefiting from large-scale Domain Generalization where all baselines fail. TDA-CLIP not only establishes a new state-of-the-art across 11 industrial and medical benchmarks but also proves its generalizability by enhancing existing SOTA methods, offering a validated and principled pathway toward building truly universal anomaly detection models.

## 1 Introduction

In critical domains such as industrial manufacturing and medical imaging, building a "universal" anomaly detection model capable of learning from diverse, multi-domain data represents a long-standing and important challenge for the field. In recent years, methods for Zero-Shot Anomaly Detection (ZS-AD) based on Vision-Language Models (VLMs) have emerged as the dominant paradigm to address this problem. This paradigm typically involves training on a source domain with supervised information (i.e., both normal and abnormal samples) to achieve zero-shot generalization on unseen target domains. Although pioneering works such as AnomalyCLIP (Zhou et al., 2024), AA-CLIP (Ma & Gao, 2025),AdaCLIP (Cao et al., 2025) and AdaptCLIP (Gao et al., 2025) have made significant progress, they are predominantly evaluated under settings with limited data diversity. Consequently, the generalization boundary of these methods when confronted with truly large-scale, cross-domain data remains underexplored.

To systematically investigate this boundary, we conducted an extensive, large-scale cross-domain training study. We trained representative state-of-the-art (SOTA) models on a dataset comprising over sixty thousand images from industrial, medical, and daily-life scenarios. The results revealed a paradoxical outcome: despite the increase in training data, all representative SOTA models suffered a marked performance decline. This evidence quantifies a sharp manifestation of the well-known challenge of negative transfer(Pan & Yang, 2009), which we term Negative Transfer in Domain Generalization(Wang et al., 2022a) in this context (illustrated in Figure 1).

Existing SOTA works have offered insightful diagnoses of this challenge from different perspectives. For instance, AnomalyCLIP attributes the issue to CLIP's inherent object-semantic bias (Zhou et al., 2024), while AA-CLIP highlights the severe overlap between normal and abnormal features in the

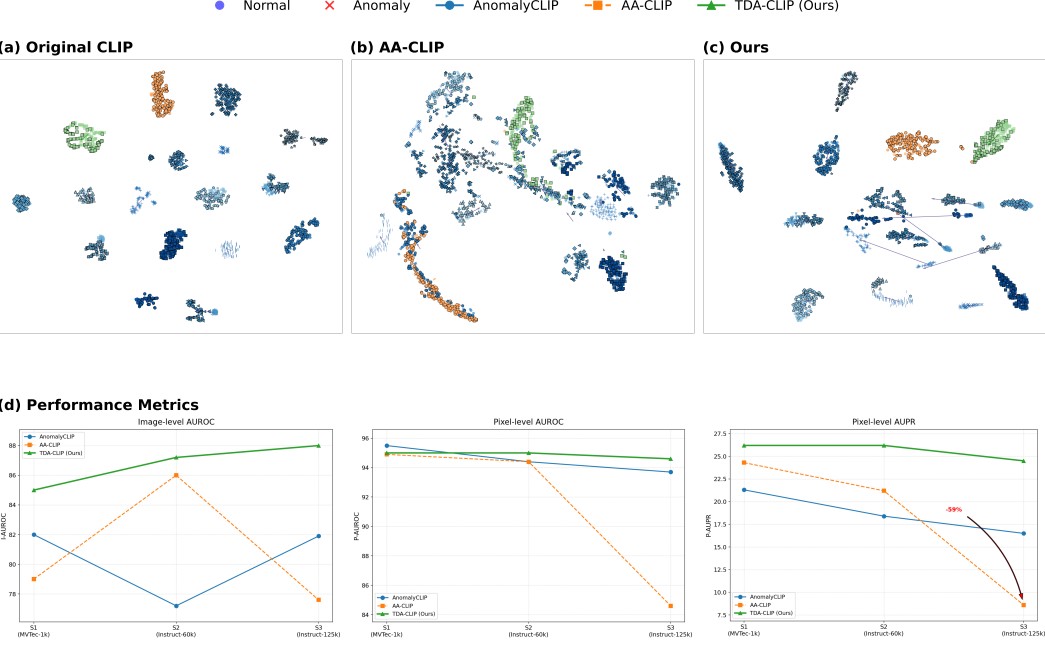

Figure 1: **Our topological framework alleviates Negative Transfer in Domain Generalization where geometric methods fail** (a) In the original CLIP feature space, normal (darker) and anomalous (lighter) samples are entangled within each class manifold. (b) After training on large-scale, diverse data, a strong geometric baseline, AA-CLIP, does not form a well-structured space, explaining its performance decline. (c) In contrast, our method (TDA-CLIP) successfully aligns the feature space, creating a consistent separation direction between normal and anomalous samples across all classes. (d) This structural difference translates to results: as data scales from S1 to S3, AA-CLIP's Pixel-AUPR decreases by 59%, while our method remains robust and benefits from data scaling.

representation space, described as "anomaly unawareness" (Ma & Gao, 2025). We argue that these explanations reveal important symptoms of the problem. Our large-scale experiments reveal that this negative transfer originates from a fundamental structural incompatibility, which we term a Domain Conflict. Prior works interpret the severe overlap between normal and anomalous features as the key obstacle, but we argue that this overlap is only a symptom. The deeper cause lies in the incompatibility of heterogeneous data manifolds: when multiple domains are embedded into a shared feature space, each imposes different and often conflicting geometric requirements. A single decision boundary cannot satisfy these heterogeneous requirements simultaneously, and this incompatibility inevitably produces the entanglement and semantic confusion observed in practice. In short, feature overlap is only a visible symptom, while the deeper root cause of failure in Domain Generalization can be understood as a Domain Conflict—the impossibility of reconciling heterogeneous manifolds with a single rigid geometry.

To address this, current methods effectively apply incremental "patches" within the geometric space. We argue that a more fundamental solution is required: a shift from enforcing rigid geometric separation toward encouraging more flexible topological separability. To this end, we propose a novel framework built on two core topological tools. First, our Homology Consistency Loss acts as a stochastic structural regularizer. Rather than attempting to recover the full dataset topology from sparse mini-batches, it enforces a simple and consistent structural principle. In practice, it aligns local feature trajectories with the global semantic direction observed across stochastic batches. Through the averaging effect of stochastic gradient descent, the model parameters converge toward a feature space that satisfies this expectation globally. Second, our Topology-Guided Attention (TGA) module serves as an auxiliary mechanism to distill high-order topological knowledge from local feature structures into the main visual encoder. Together, these components achieve a robust macro–micro synergy.

In summary, our contributions are as follows:

1. We systematically quantify the severe impact of Negative Transfer in Domain Generalization on SOTA ZS-AD methods, and we introduce the concept of a Domain Conflict as a unifying explanation of its structural root cause.

2. We propose a topology-based framework, TDA-CLIP, which resolves this robustness bottleneck by shifting the objective from geometric separation to topological separability, enforced via our stochastic structural regularizer.

3. Extensive experiments demonstrate that our method not only surpasses prior SOTA on standard benchmarks but also, importantly, remains robust and benefits from large-scale Domain Generalization, where all tested baselines degrade.

## 2 RELATED WORK

### 2.1 CLIP-BASED ANOMALY DETECTION

The application of CLIP to anomaly detection has recently gained significant attention. Early works focused on using pre-trained CLIP features in a zero-shot or few-shot manner (Jeong et al., 2023). More advanced methods introduce learnable components, such as visual prompts (Jia et al., 2022; Li et al., 2024) or lightweight adapters (Gao et al., 2024; 2025; Huang et al., 2024; Qu et al., 2024), to better adapt the model to the anomaly detection task. A common thread across these works is their reliance on a loss function that minimizes the geometric distance (e.g., cosine or L2 distance) between image patch embeddings and corresponding textual concept embeddings "normal" vs. "anomaly"). While effective in single-domain settings, we argue that this fundamental reliance on geometric similarity is the primary obstacle to successful large-scale, multi-domain training.

### 2.2 DOMAIN GENERALIZATION AND NEGATIVE TRANSFER

The challenge of learning from multiple source domains to generalize to unseen targets has long been studied under the umbrella of Domain Generalization (Wang et al., 2022b). A related and well-known issue is negative transfer, where incorporating additional source-domain data inadvertently harms generalization to target domains (Zhang et al., 2022; Li et al., 2025b). Recent analyses attribute such degradation to conflicting optimization signals or domain-specific gradients during training, where updates that benefit one domain may deteriorate performance on another (Li et al., 2025a). However, most existing works analyze this issue at the inter-domain level. In contrast, its manifestation in large-scale, diverse training for zero-shot anomaly detection (ZS-AD)—where performance can even degrade on domains already present in training—remains underexplored.

### 2.3 TOPOLOGICAL DATA ANALYSIS IN DEEP LEARNING

Topological Data Analysis (TDA), particularly persistent homology, has become a powerful tool for analyzing high-dimensional data structures in machine learning (Dey & Wang, 2022). Its applications are diverse, ranging from enforcing topological priors in computer vision tasks like segmentation (Hu et al., 2019; Clough et al., 2020) to enhancing graph neural networks (Yan et al., 2021; Immonen et al., 2023).

A significant line of work uses TDA to regularize and analyze representation spaces. These methods typically aim to *preserve* the topology of the input data during dimensionality reduction (Moor et al., 2020), or maintain the structural integrity of pre-trained models during fine-tuning. For instance, a recent work (Zhang et al., 2024) utilizes a homology consistency loss to ensure that the overall topological signatures of image and text manifolds remain aligned.

In sharp contrast, our work represents a paradigm shift. We do not use TDA as a preservative regularizer to maintain the status quo. Instead, we elevate it to a core optimization objective for actively shaping the structure of the feature space. Our loss function is fundamentally different: it identifies and aligns the topological critical paths connecting normal and anomalous samples with a global semantic direction. By doing so, our method directly constructs a topologically separable feature space to solve the critical problem of negative transfer, rather than merely preserving an existing one.

## 3 METHODOLOGY

To fundamentally resolve the "Domain Conflict", we propose TDA-CLIP, a framework centered on a novel topologically-guided training strategy. The overall architecture of our framework is illustrated in Figure 2. Our approach is built upon a decoupled two-stage training strategy that synergistically integrates macro-level global constraints and micro-level local refinements to instill topological robustness into the model. This strategy aims to distill high-order topological structural constraints into the parameters of the main visual encoder via an auxiliary module (TGA) that is entirely pruned at inference time. The result is an encoder with high robustness to Negative Transfer in Domain Generalization, which significantly enhances anomaly detection performance without introducing any additional inference cost, as depicted in Figure 2(d).

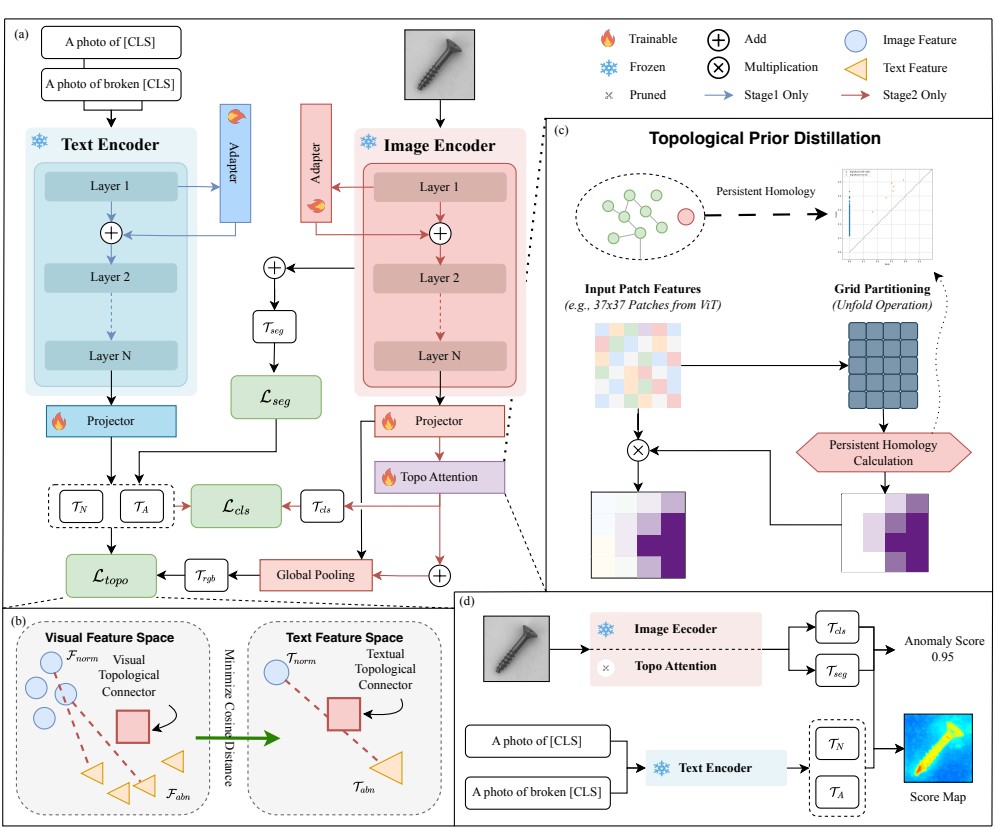

Figure 2: **The overall architecture of our TDA-CLIP framework (a)** The main two-stage training architecture, showing the data flow for the text and image encoders augmented with lightweight adapters. **(b)** A conceptual visualization of our macro-level Homology Consistency Loss ($\mathcal{L}_{HC}$), which aims to minimize the cosine distance between the Visual and Textual Topological Connectors. **(c)** A detailed illustration of the training-only Topology-Guided Attention (TGA) module. It takes patch features as input and generates a topo-attention map by calculating persistent homology on local grids. This corresponds to the "Topo-attention" block in (a). **(d)** The efficient inference pipeline, where the entire TGA module is pruned, resulting in zero additional computational cost .

### 3.1 MACRO-LEVEL: GLOBAL STRUCTURE SHAPING VIA $\mathcal{L}_{HC}$

At the macro-level, we address the challenge of how to sculpt the manifolds in the global feature space $\mathcal{V} \subset \mathbb{R}^D$ to disentangle the entangled feature clusters of the normal class, $\mathcal{M}_{norm}$, and the anomalous class, $\mathcal{M}_{anom}$. Our solution is the Anomaly-Aware Homology Consistency Loss ($\mathcal{L}_{HC}$),

a top-down regularization tool designed to enforce a global structural principle. The core objective of this loss is conceptually visualized in Figure 2(b).

To formally define this loss, we employ Persistent Homology. For a mini-batch of $B$ images, we first extract their image-level feature vectors to form a point cloud $Z = \{z_i\}_{i=1}^{B}$. We then apply 0-dimensional persistent homology by building a filtration $\mathcal{F}$ over this point cloud. This process tracks how connected components merge and identifies the set of all *0-dimensional death edges*, denoted $E_{\text{death}}(\mathcal{F})$. An edge $[z_i, z_j] \in E_{\text{death}}(\mathcal{F})$ represents a critical topological bridge that connects two previously distinct components of the data. The detailed computational method is described in Algorithm 4 and Appendix E.1.

We hypothesize that these persistence-critical pairs offer a more robust signal for global separation than traditional geometric hard-mining, which often focuses on noisy local boundaries instead of structurally significant connections. This is empirically validated in Section 4.2.3.

From the set of death edges, we filter for the pairs that are most informative for our task: those connecting normal and anomalous samples. We define the set of Persistence-Critical Pairs $\mathcal{P}_{\text{critical}}$ as:

$$\mathcal{P}_{\text{critical}} = \{(i,j) \mid [z_i, z_j] \in E_{\text{death}}(\mathcal{F}) \land y_i \neq y_j\} \tag{1}$$

where $i$ and $j$ are sample indices from the batch, and $y_i \in \{0, 1\}$ is the class label of feature $z_i$. The objective of $\mathcal{L}_{\text{HC}}$ is to align the direction of these critical paths with the global semantic discriminant vector $w = t_{\text{anom}} - t_{\text{norm}}$, where $t_{\text{anom}}, t_{\text{norm}} \in \mathbb{R}^D$ are the textual embeddings for "anomaly" and "normal." For every $(i, j) \in \mathcal{P}_{\text{critical}}$, we define a Visual Topological Connector $v_{ij} = z_j - z_i$, oriented from the normal to the anomalous sample. The loss then maximizes the directional consistency between $v_{ij}$ and the Textual Topological Connector $w$:

$$\mathcal{L}_{\text{HC}} = \mathbb{E}_{(i,j)\in\mathcal{P}_{\text{critical}}} \left[ 1 - \frac{\langle v_{ij}, w \rangle}{\|v_{ij}\|_2 \cdot \|w\|_2} \right] \tag{2}$$

where $\langle \cdot, \cdot \rangle$ denotes the inner product. It is important to note that $\mathcal{P}_{\text{critical}}$ may be empty for mini-batches containing samples of only a single class. In such cases, the loss term $\mathcal{L}_{\text{HC}}$ for that batch is simply zero. The overall effectiveness of this regularization is achieved through stochastic averaging over numerous diverse mini-batches throughout the training process. As shown in our mathematical proof (see Appendix E.2), minimizing this loss implicitly optimizes for geometric separation while regularizing the topology.

### 3.2 Micro-level: Local Evidence Refinement via TGA

While the macro-level constraint provides a top-down design, its effectiveness hinges on the quality of the local visual evidence. A bottom-up refinement mechanism is therefore indispensable. To this end, we designed the Topology-Guided Attention (TGA) module, a training-only component that acts as a feature purifier at the micro-scale. The design is based on the core Topological Outlier Hypothesis: in a well-formed feature space, anomalous patch features should be topologically isolated from the "mainland" of normal patch features within a local neighborhood, as visually demonstrated in Figure 3.

Let the batch of patch features from the visual encoder be represented as $F_{\text{patch}} = \{f_i\}_{i=1}^{N}$ for a single image, where $N = H \times W$ is the number of patches and $D$ is the feature dimension. The TGA process can be decomposed into the following steps:

**1. Local Patch Partitioning** To analyze local topology, we partition the grid of patch indices $\mathcal{I} = \{1, \ldots, N\}$ into $k^2$ disjoint subsets $\mathcal{I}_{ab}$, where $(a, b)$ denotes the cell coordinates. This partitioning, efficiently implemented via an 'unfold' operation, creates $k^2$ local point clouds for parallel analysis. Each point cloud $\mathcal{M}_{ab}$ is defined as the set of patch features within its corresponding cell:

$$\mathcal{M}_{ab} = \{f_i \mid i \in \mathcal{I}_{ab}\} \tag{3}$$

**2. Parallel Topological Saliency Computation** For each local point cloud $\mathcal{M}_{ab}$, we compute its 0-dimensional death edges, $E_{\text{death}}(\mathcal{M}_{ab})$, in parallel using the principles described in Section 3.1. Motivated by our Topological Outlier Hypothesis (Figure 3), we define the saliency of a patch feature $f_i$ as the sum of the death times ($r_{\text{death}}$) of all death edges connected to it. A high saliency score

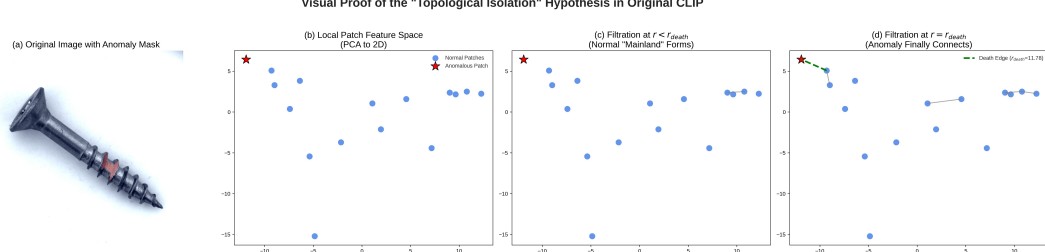

Figure 3: **Visual proof of the Topological Outlier Hypothesis** (a) An anomalous screw image with the defect highlighted. (b) The 2D projection of local patch features around the anomaly, where the anomalous patch (red star) is distinct from normal patches (blue circles). (c) As the connection radius $r$ in the Vietoris-Rips filtration increases, normal patches connect to form a "mainland". The anomalous patch remains isolated. (d) Only at a large radius, the death time $r_{\text{death}}$, does the anomalous patch finally connect to the mainland via the "death edge" (green dashed line). This large connection distance is a measure of its topological saliency.

indicates that a patch remains topologically isolated for longer, suggesting it is anomalous. The score is computed as:

$$s(\boldsymbol{f}_i) = \sum_{e=[\boldsymbol{f}_i, \boldsymbol{f}_j] \in E_{\text{death}}(\mathcal{M}_{ab})} r_{\text{death}}(e) \tag{4}$$

This process yields a saliency score for every patch feature across the image.

**3. Attention-based Feature Aggregation** The patch saliency scores are reassembled into a map $S \in \mathbb{R}^{H \times W}$, which is then converted into a spatial attention map $A$ via a softmax function with temperature $\tau$:

$$A_i = \frac{\exp(S_i/\tau)}{\sum_{j=1}^{N} \exp(S_j/\tau)} \tag{5}$$

The attention map is used to compute a weighted aggregation of the original patch features, producing a single, topology-refined feature $\boldsymbol{z}_{\text{tda}}$:

$$\boldsymbol{z}_{\text{tda}} = \sum_{i=1}^{N} A_i \cdot \boldsymbol{f}_i \tag{6}$$

This feature encapsulates the most topologically salient information and serves as a "teacher" signal for the subsequent training stage.

### 3.3 DECOUPLED TWO-STAGE TRAINING: A MACRO-MICRO SYNERGY

To stably integrate our macro-level global shaping and micro-level local refinement objectives, we design a decoupled two-stage training strategy. This approach avoids conflicting gradients from simultaneous optimization. Following established practices (Gao et al., 2024), our training exclusively fine-tunes lightweight Adapter modules while keeping the original CLIP encoders frozen. The data flow is orchestrated in Figure 2(a).

#### 3.3.1 STAGE 1: SEMANTIC ANCHOR LEARNING

The objective of this stage is to learn a stable semantic discriminant vector $\boldsymbol{w} = \boldsymbol{t}_{\text{anom}} - \boldsymbol{t}_{\text{norm}}$. We freeze the visual encoder and train only the Adapters in the text encoder. The optimization objective is $\mathcal{L}_{\text{Stage1}} = \mathcal{L}_{\text{seg}} + \mathcal{L}_{\text{HC}}$. Crucially, to provide a stable topological target, the features for both losses are sourced from the original, frozen visual encoder:For $\mathcal{L}_{\text{seg}}$ (Focal Loss): Computed on patch features $\boldsymbol{F}_{\text{patch}}$ from the frozen encoder.For $\mathcal{L}_{\text{HC}}$: The point cloud $Z$ is constructed from image-level features $\boldsymbol{z}_{\text{avg}}$ (global average pooled) from the frozen encoder. This setup ensures the learned semantic anchor $\boldsymbol{w}$ aligns with the well-structured feature space of the pre-trained CLIP model.

### 3.3.2 STAGE 2: TOPOLOGY-AWARE VISUAL ADAPTATION

With a stable semantic anchor, this stage reshapes the visual feature space. We freeze the optimized text encoder and train the Adapters of the visual encoder, guided by the TGA module in a knowledge distillation paradigm. The TGA ("teacher") produces a topology-refined feature $z_{\text{tda}}$ from the student's patch features. To distill topological knowledge into the student's main feature representation, $z_{\text{avg}}$ (global average pooled feature), we construct a temporary, fused feature for supervision:

$$z_{\text{out}} = z_{\text{avg}} + \alpha \cdot z_{\text{tda}} \tag{7}$$

The multi-task loss is then defined as:

$$\mathcal{L}_{\text{Stage2}} = \lambda_{\text{seg}}\mathcal{L}_{\text{seg}} + \lambda_{\text{cls}}\mathcal{L}_{\text{cls}} + \lambda_{\text{align}}\mathcal{L}_{\text{align}} + \lambda_{\text{hc}}\mathcal{L}_{\text{HC}} \tag{8}$$

The loss components are precisely targeted. The primary task losses:a pixel-wise Focal Loss ($\mathcal{L}_{\text{seg}}$) on $F_{\text{patch}}$ and an image-level Dice Loss ($\mathcal{L}_{\text{cls}}$) on the student's class token feature $z_{\text{cls}}$ ground the student in the main objectives. The auxiliary topological losses are both applied to the teacher-guided fused feature $z_{\text{out}}$ from Eq. 7. Specifically, $\mathcal{L}_{\text{align}}$ is an image-level Dice Loss that ensures the fused feature is semantically classifiable, while for $\mathcal{L}_{\text{HC}}$, the point cloud $Z$ is constructed from the set of $z_{\text{out}}$ vectors to enforce global structural consistency. This co-optimization compels the student (via backpropagation through $z_{\text{avg}}$) to internalize the robust topological structures guided by the teacher.

## 4 EXPERIMENTS

### 4.1 EXPERIMENTAL SETUP

To systematically evaluate our method's ability to overcome Negative Transfer in Domain Generalization, we conducted extensive experiments on a wide range of industrial and medical anomaly detection benchmarks. Our core evaluation strategy involves training and comparing models across three progressively scaled settings: **S1** (single-domain), **S2** (multi-domain, clean), and **S3** (large-scale, heterogeneous), allowing us to directly measure performance as data diversity increases. For a fair comparison, all methods were benchmarked against strong state-of-the-art baselines under strictly controlled conditions. The complete details of our datasets, training protocols, baseline implementations, and evaluation metrics are provided in the Appendix F.1 and F.2

### 4.2 MAIN RESULTS AND ANALYSES

#### 4.2.1 SOTA COMPARISON ON STANDARD BENCHMARKS (S1 SETTING)

We first validate the performance of our method in the standard S1 single-domain generalization setting. As shown in Table 1, our method achieves leading performance across multiple datasets. For instance, on the industrial dataset average, our TDA-CLIP achieves an Image-AUROC of **84.6%** and a Pixel-AUPR of **37.2%**, which is the most challenging metric, surpassing all prior state-of-the-art methods. This result strongly demonstrates that even in a standard setting with limited data scale, our topological constraint approach can effectively enhance the model's feature representation and generalization capabilities.

#### 4.2.2 OVERCOMING NEGATIVE TRANSFER IN DOMAIN GENERALIZATION IN LARGE-SCALE SETTINGS

The core motivation of our work is most directly validated in Figure 1(d) and Table 2. As training data scales from S1 to S3, baseline methods suffer a sharp, cross-domain performance collapse due to the "Domain Conflict", with AA-CLIP's average Pixel-AUPR collapsing by over 59%. In stark contrast, our TDA-CLIP is the only method to remain robust and consistently benefit from additional data, with its Image-AUROC on VisA steadily improving to a peak of 88.0% in the S3 setting. This result irrefutably proves that our topological paradigm successfully escapes the "Domain Conflict", offering a truly viable path for building large-scale, general-purpose anomaly detection models.

Table 1: Detailed performance comparison across all 11 industrial and medical datasets under the S1 training setting. We report Pixel-AUROC / Pixel-AUPR and Image-AUROC (%). Best in **bold**, second-best is underlined.

| Dataset | WinCLIP | AdaCLIP | AnomalyCLIP | AA-CLIP | TDA-CLIP |
|---|---|---|---|---|---|
| **Pixel-AUROC / Pixel-AUPR** | | | | | |
| MVTec AD | 82.3 / 18.2 | 88.8 / 40.4 | 91.1 / 34.5 | 91.9 / 45.3 | **92.2** / **48.6** |
| VisA | 73.2 / 5.4 | **95.7** / **31.0** | 95.5 / 21.3 | 94.9 / 24.3 | 95.0 / 26.2 |
| Real-IAD | 84.5 / 3.3 | **96.1** / **30.5** | 95.1 / 26.7 | 95.2 / 29.1 | 95.6 / 30.4 |
| MPDD | 71.2 / 14.1 | 95.5 / 25.9 | 96.2 / **28.0** | **96.7** / 26.8 | **96.7** / 26.9 |
| BTAD | 72.7 / 12.9 | 94.2 / 48.0 | 94.2 / 45.5 | 97.0 / 52.8 | **97.5** / **53.7** |
| *Industrial Mean* | *76.8 / 10.8* | *94.1 / 35.2* | *94.4 / 31.2* | *95.1 / 35.7* | *95.4 / 37.2* |
| ColoDB | 51.2 / 14.3 | 76.6 / 27.2 | 82.9 / 34.3 | 84.0 / 33.6 | **85.4** / **35.6** |
| ClinicDB | 70.3 / 19.4 | 83.6 / 43.7 | 85.0 / 41.8 | 89.0 / 52.6 | **91.0** / **57.0** |
| CVC-300 | 88.2 / 20.6 | 89.1 / 32.1 | 95.4 / 55.3 | 96.4 / 53.7 | **97.0** / **59.3** |
| Kvasir | 69.7 / 27.8 | 78.2 / 45.6 | 81.9 / 42.5 | 87.2 / 55.8 | **88.0** / **57.8** |
| Retina | 80.6 / 43.8 | 91.5 / 59.3 | 93.3 / 62.9 | 95.5 / 50.8 | **96.0** / **68.2** |
| Brain | 86.0 / 21.2 | 94.2 / 39.5 | 96.2 / 33.3 | 95.5 / 44.9 | **96.0** / **46.2** |
| *Medical Mean* | *74.3 / 24.5* | *85.5 / 41.2* | *89.1 / 45.0* | *90.8 / 48.6* | *93.3 / 54.0* |
| **Overall Mean** | 75.4 / 18.3 | 89.4 / 38.5 | 91.5 / 38.7 | 93.0 / 42.7 | **93.7** / **46.3** |
| **Image-AUROC** | | | | | |
| MVTec AD | 90.4 | 90.7 | 91.6 | 90.5 | **93.2** |
| VisA | 75.5 | 81.7 | 82.0 | 79.0 | **85.0** |
| Real-IAD | 67.0 | 73.3 | 69.5 | 76.0 | **76.6** |
| MPDD | 61.5 | 72.7 | **73.6** | 75.1 | 72.7 |
| BTAD | 68.2 | 89.6 | 89.1 | 94.8 | **95.3** |
| *Industrial Mean* | *72.5* | *81.6* | *81.2* | *83.1* | *84.6* |
| Retina | 42.5 | 82.7 | 75.7 | 82.7 | **85.5** |
| Brain | 66.5 | 80.2 | **83.3** | 80.2 | 81.2 |
| *Medical Mean* | *54.5* | *81.5* | *79.5* | *81.5* | *83.4* |
| **Overall Mean** | 67.4 | 81.6 | 80.7 | 82.6 | **84.2** |

Table 2: Performance evolution on VisA (industrial) and Kvasir (medical) datasets when training on datasets of increasing scale and diversity. For each method, the best performance across the three settings is highlighted in bold, illustrating that only our TDA-CLIP consistently benefits from more data. The "Mean" column averages the available pixel-level metrics.

| Method | Training Set | VisA | | | Kvasir | | Mean | |
|---|---|---|---|---|---|---|---|---|
| | | Image-AUROC | Pixel-AUROC | Pixel-AUPR | Pixel-AUROC | Pixel-AUPR | Pixel-AUROC | Pixel-AUPR |
| AnomalyCLIP | MVTec-1k (S1) | **82.0** | **95.5** | **21.3** | **81.9** | **39.6** | **88.7** | **30.5** |
| | Instruct-40k (S2) | 77.2 | 94.4 | 18.4 | 79.3 | 39.1 | 86.8 | 28.8 |
| | Instruct-125k (S3) | 81.9 | 93.7 | 16.5 | 78.0 | 36.8 | 85.9 | 26.7 |
| AA-CLIP | MVTec-1k (S1) | 79.0 | **94.9** | **24.3** | **83.3** | **45.5** | **89.1** | **34.9** |
| | Instruct-40k (S2) | **86.0** | 94.4 | 21.2 | 74.6 | 32.9 | 84.5 | 27.1 |
| | Instruct-125k (S3) | 77.6 | 84.6 | 8.6 | 64.0 | 19.9 | 74.3 | 14.3 |
| **TDA-CLIP** | MVTec-1k (S1) | 85.0 | 95.0 | 26.2 | **81.8** | 41.5 | **88.4** | 33.8 |
| | Instruct-40k (S2) | 87.2 | **95.0** | **26.2** | 81.5 | 41.9 | 88.3 | 34.1 |
| | Instruct-125k (S3) | **88.0** | 94.6 | 24.5 | 81.3 | **43.9** | 88.0 | **34.2** |

### 4.2.3 ON THE SUPERIORITY OF TOPOLOGICAL PAIR MINING

To empirically validate our central hypothesis that topological pair mining is superior to geometric hard-mining, we conducted a direct comparison, with results summarized in Table 3. The outcome is unequivocal: our method of selecting pairs based on "topological criticality" significantly outperforms strategies based on geometric distance or loss values. This result confirms that the "bridging pairs" identified by persistent homology (visualized in Appendix H.2) are indeed more structurally

informative for learning a robustly separable feature space than traditional "hard pairs", validating the core motivation for our framework's design.

Table 3: Comparison of different pair mining strategies on the Industrial Mean performance, when combined with the same directional consistency loss.

| Pair Mining Strategy | Pixel-AUPR | Pixel-AUROC | Image-AUROC |
|---|---|---|---|
| Random Pairs | 36.1 | 94.2 | 81.7 |
| Nearest-Neighbor Hard Mining | 36.3 | 95.1 | 83.1 |
| Max-Loss Hard Mining | 36.3 | 95.2 | 82.9 |
| **Persistence-Critical Pairs (Ours)** | **36.8** | **95.3** | **84.3** |

#### 4.2.4 ABLATION STUDY: DISSECTING THE TOPOLOGICAL COMPONENTS

Table 4: Ablation study of our components on the Industrial Mean performance. Our baseline consists of the same adapter-based architecture trained solely with a standard cross-entropy loss and a pixel-wise segmentation loss, without our proposed topological components.

| Method | Pixel-AUPR | Pixel-AUROC | Image-AUROC |
|---|---|---|---|
| Baseline | 35.1 | 94.9 | 82.2 |
| + $\mathcal{L}_{\text{HC}}$ (Stage 1 & 2) | 36.8 | 95.3 | 84.3 |
| + TGA(Stage 2 only) | 36.1 | 95.0 | 83.6 |
| + Both (TDA-CLIP) | **37.2** | **95.4** | **84.6** |

To understand the individual contributions of our components, we conducted an ablation study summarized in Table 4. The results clearly demonstrate a powerful synergistic effect. The baseline model achieves an industrial mean Pixel-AUPR of 35.1%. Adding only the macro-level constraint $\mathcal{L}_{\text{HC}}$ significantly boosts this to 36.8% (+1.7%), validating the effectiveness of global topological regularization. Separately, adding only the micro-level TGA module improves performance to 36.1% (+1.0%), highlighting the importance of local feature purification. When both components are combined in our final TDA-CLIP model, performance reaches a peak of 37.2% (+2.1%), an improvement greater than the sum of the individual parts, confirming the strong synergy between our macro- and micro-level tools.

### 4.3 COMPUTATIONAL EFFICIENCY ANALYSIS

Our framework is computationally efficient. Crucially, all topological modules are pruned at inference time, resulting in zero additional cost compared to the baseline. The training overhead is also minimal; due to efficient, parallelized computations, the total training time is only about 20% longer, which is a modest trade-off for the significant robustness gains achieved.

## 5 CONCLUSION

In this work, we identified and tackled the key bottleneck of universal anomaly detection: Negative Transfer in Domain Generalization. We diagnose its root cause as the over-reliance on rigid geometric separation, a form of metric overfitting, which in the context of Domain Generalization manifests as what we term a Domain Conflict. To overcome this, we introduced topological separability through two powerful, inference-free tools: the Homology Consistency Loss and Topology-Guided Attention. While the computation of persistent homology introduces a modest overhead during training compared to prior paradigms, this is a worthwhile trade-off. Our framework not only avoids performance collapse under large-scale Domain Generalization but also enhances existing methods, delivering a reusable and principled toolkit for building robust vision models at scale with zero additional inference cost.

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

## A    Appendix

## B    Ethics Statement

This work adheres to the ICLR Code of Ethics. In this study, no human subjects or animal experimentation were involved. All datasets used, including MVTec-AD, VisA, Real-IAD, BTAD, MPDD (industrial benchmarks), and ColoDB, ClinicDB, CVC-300, Kvasir, Retina, Brain (medical benchmarks), were sourced in compliance with their respective usage licenses and guidelines, ensuring no violation of privacy. We have taken care to avoid any biases or discriminatory outcomes in our research process. No personally identifiable information was used, and no experiments were conducted that could raise privacy or security concerns. We are committed to maintaining transparency and integrity throughout the research process.

## C    Reproducibility Statement

We have made every effort to ensure that the results presented in this paper are reproducible. All code and datasets have been made publicly available in an anonymous repository to facilitate replication and verification. The experimental setup, including training steps, model configurations, and hardware details, is described in detail in the paper and appendix. We have also provided a full description of our contribution, including the design and implementation of Homology Consistency Loss and Topology-Guided Attention, to assist others in reproducing our experiments.

Additionally, the public datasets we used, including industrial benchmarks (MVTec-AD, VisA, Real-IAD, BTAD, MPDD) and medical benchmarks (ColoDB, ClinicDB, CVC-300, Kvasir, Retina, Brain), are all publicly available, ensuring consistent and reproducible evaluation results.

We believe these measures will enable other researchers to reproduce our work and further advance the field.

## D    LLM Usage

Large Language Models (LLMs) were used to aid in the writing and polishing of the manuscript. Specifically, we used an LLM to assist in refining the language, improving readability, and ensuring clarity in various sections of the paper. The model helped with tasks such as sentence rephrasing, grammar checking, and enhancing the overall flow of the text.

It is important to note that the LLM was not involved in the ideation, research methodology, or experimental design. All research concepts, ideas, and analyses were developed and conducted by the authors. The contributions of the LLM were solely focused on improving the linguistic quality of the paper, with no involvement in the scientific content or data analysis.

The authors take full responsibility for the content of the manuscript, including any text generated or polished by the LLM. We have ensured that the LLM-generated text adheres to ethical guidelines and does not contribute to plagiarism or scientific misconduct.

# E   METHODOLOGICAL DETAILS

## E.1   COMPUTATIONAL DETAILS OF PERSISTENT HOMOLOGY

This appendix provides a brief technical overview of the computational pipeline used to identify the 0-dimensional *death edges* for our Homology Consistency Loss ($\mathcal{L}_{\text{HC}}$) and Topology-Guided Attention (TGA) module. Our approach is based on constructing a Vietoris-Rips filtration over a point cloud of features.

**1. Point Cloud and Distance Metric**   Let $Z = \{z_i\}_{i=1}^{B}$ be a point cloud of $B$ feature vectors in $\mathbb{R}^D$. The fundamental input for building a filtration is the pairwise distance between all points. We use the standard Euclidean distance ($L_2$ norm) to define the distance $d(z_i, z_j) = \|z_i - z_j\|_2$.

**2. Vietoris-Rips (VR) Filtration**   A filtration is a sequence of nested topological spaces. The Vietoris-Rips filtration, denoted $\mathcal{F}_{\text{VR}}$, is a standard and computationally efficient method for constructing such a sequence from a point cloud. It is built upon the concept of a **simplicial complex**, which is a collection of vertices, edges, triangles, tetrahedra, and their higher-dimensional counterparts.

The VR complex at a given radius $r \geq 0$, denoted $\text{VR}(Z, r)$, is a simplicial complex whose vertices are the points in $Z$. A set of $k+1$ vertices $\{z_{i_0}, \ldots, z_{i_k}\}$ forms a $k$-simplex in $\text{VR}(Z, r)$ if and only if the pairwise distance between any two vertices in the set is at most $r$.

- **0-simplices:** The points $\{z_i\}$ themselves.
- **1-simplices (Edges):** A pair $[z_i, z_j]$ is an edge if $d(z_i, z_j) \leq r$.
- **2-simplices (Triangles):** A triplet $[z_i, z_j, z_k]$ is a triangle if all three of its edges have length at most $r$.

The filtration $\mathcal{F}_{\text{VR}}$ is the sequence of VR complexes obtained by continuously increasing the radius $r$ from 0 to infinity:

$$\text{VR}(Z, r_1) \subseteq \text{VR}(Z, r_2) \quad \text{for any } 0 \leq r_1 \leq r_2$$

As $r$ increases, new simplices (edges, triangles, etc.) are added to the complex, causing topological features like connected components, loops, and voids to appear and disappear.

**3. 0-Dimensional Persistent Homology and Death Edges**   Persistent homology is an algebraic method for tracking the "lifespan" of these topological features across the filtration. We focus on **0-dimensional homology** ($H_0$), which counts the number of connected components in the complex.

- At $r = 0$, each point is its own connected component, so there are $B$ components.
- As $r$ increases, edges are added. When an edge $[z_i, z_j]$ connects two previously separate connected components, one component "dies" and merges into the other. The number of connected components decreases by one.
- The radius $r$ at which this merge occurs is called the **death time** of the dying component. The edge that causes this merge is the corresponding **0-dimensional death edge**.

The set $E_{\text{death}}(\mathcal{F})$ used in our main paper is precisely the set of all such edges that reduce the number of connected components during the filtration. Algorithm 4 provides an efficient implementation for finding these edges using a Union-Find data structure, which effectively simulates this process without needing to construct the full simplicial complexes. Each death edge represents a critical topological link that connects major structural parts of the point cloud, making it an ideal candidate for our topologically guided loss.

## E.2   MATHEMATICAL RATIONALE: WHY TOPOLOGICAL CONSISTENCY OPTIMIZES GEOMETRIC SEPARATION

This section provides a formal mathematical argument for why our novel training objective, the Homology Consistency Loss ($\mathcal{L}_{\text{HC}}$), is not only theoretically sound but also inherently optimizes the conventional geometric separation test, leading to its powerful generalization capabilities.

### E.2.1 THE MATHEMATICAL ESSENCE OF THE GEOMETRIC TEST: HYPERPLANE SEPARATION

First, we formalize the standard anomaly detection test. Let $z_i \in \mathbb{R}^D$ be the image-level feature vector of a test image $i$, and let $t_{\text{norm}}, t_{\text{anom}} \in \mathbb{R}^D$ be the text feature vectors for the concepts "normal" and "anomaly," respectively. The test score $S(i)$ is typically computed as:

$$S(i) = \text{sim}(z_i, t_{\text{anom}}) - \text{sim}(z_i, t_{\text{norm}}) \tag{9}$$

Assuming normalized features and cosine similarity, this is equivalent to:

$$S(i) = z_i \cdot t_{\text{anom}} - z_i \cdot t_{\text{norm}} = z_i \cdot (t_{\text{anom}} - t_{\text{norm}}) \tag{10}$$

We define a fixed semantic discriminant vector $w = t_{\text{anom}} - t_{\text{norm}}$, which points from "normal" to "anomaly" in the semantic space. The test score simplifies to the projection of the image feature onto this semantic vector:

$$S(i) = z_i \cdot w \tag{11}$$

From a geometric perspective, the goal of a conventionally trained model is to learn a feature space where a hyperplane, with normal vector $w$, can effectively separate the feature points of normal samples from those of anomalous samples. The training objective is to directly maximize the margin between the two classes with respect to this hyperplane.

### E.2.2 THE MATHEMATICAL ESSENCE OF THE TOPOLOGICAL OBJECTIVE: LOCAL DIRECTIONAL ALIGNMENT

Our Homology Consistency Loss ($\mathcal{L}_{\text{HC}}$) adopts a different, more fundamental approach. Instead of directly optimizing a global hyperplane, $\mathcal{L}_{\text{HC}}$ imposes a stricter, fine-grained topological constraint on "critical paths" connecting normal and anomalous samples. These paths are identified via persistent homology. For a critical pair of samples $(i, j)$ where sample $i$ is normal and sample $j$ is anomalous, we define two key vectors. The Visual Trajectory is defined as $v_{ij} = z_j - z_i$, while the Semantic Trajectory is simply the global semantic vector $w$, as it represents the conceptual path from "normal" to "anomaly." The loss function aims to enforce directional alignment between these two vectors by minimizing their cosine distance:

$$\mathcal{L}_{\text{HC}} = \mathbb{E}_{(i,j) \in \mathcal{P}_{\text{critical}}} \left[ 1 - \cos(v_{ij}, w) \right] = \mathbb{E}_{(i,j) \in \mathcal{P}_{\text{critical}}} \left[ 1 - \frac{(z_j - z_i) \cdot w}{|z_j - z_i|_2 \cdot |w|_2} \right] \tag{12}$$

where $\mathcal{P}_{\text{critical}}$ is the set of all critical cross-class pairs.

### E.2.3 THE MATHEMATICAL BRIDGE: HOW TOPOLOGICAL CONSTRAINTS OPTIMIZE GEOMETRIC SEPARATION

Minimizing $\mathcal{L}_{\text{HC}}$ forces the model to make the visual displacement vector, $z_j - z_i$, for all structurally critical pairs $(i, j)$, parallel and co-directional with the global semantic discriminant vector $w$. This seemingly local constraint has powerful mathematical consequences.

This leads to a transition from local alignment to global order. While the geometric test only requires that the projection of $z_j$ onto $w$ is greater than that of $z_i$, our topological loss imposes a much stricter condition: the vector $(z_j - z_i)$ itself must align with $w$. By enforcing this alignment consistently for all critical paths across the entire dataset, we are effectively "combing" the feature space. This process regularizes the space into a globally ordered structure, where all paths from normal to anomalous regions point in a unified direction, preventing the feature space from twisting or entangling when fitting complex, multi-domain data.

Furthermore, this objective leads to an implicit margin maximization. Let's rewrite the core term in the numerator of $\mathcal{L}_{\text{HC}}$:

$$(z_j - z_i) \cdot w = z_j \cdot w - z_i \cdot w = S(j) - S(i) \tag{13}$$

Minimizing $\mathcal{L}_{\text{HC}}$ is equivalent to maximizing the cosine term. Assuming the norms $|z_j - z_i|_2$ and $|w|_2$ are relatively stable (which is encouraged by feature normalization), this maximization is dominated by the need to maximize the dot product $(z_j - z_i) \cdot w$. This is precisely an objective to maximize the margin, $S(j) - S(i)$, between the test scores of anomalous and normal samples.

This reveals a key distinction. Geometric methods seek a "loose" global separation on all points, which can fail under data complexity. In contrast, our topological method imposes an "extremely strict" local directional constraint on the most structurally critical pairs. This strict local constraint, through its global consistency (all paths align with the same $\boldsymbol{w}$), gives rise to an emergent, highly-ordered global feature space that is easily separable by a hyperplane.

### E.2.4 The Source of Generalization: Learning a Principle, Not an Appearance

The powerful generalization of our model stems from its ability to learn a domain-agnostic, abstract structural principle, rather than specific geometric features of objects.

The model learns a feature representation that satisfies a topo-semantic consistency constraint. For all critical pairs $(i, j)$ in the training set, the process of "becoming anomalous" in the image space is mapped to a displacement in the feature space that is parallel to the corresponding semantic displacement:

$$\forall (i, j) \in \mathcal{P}_{\text{critical}}, \quad (\boldsymbol{z}_j - \boldsymbol{z}_i) \parallel (\boldsymbol{t}_{\text{anom}} - \boldsymbol{t}_{\text{norm}}) \tag{14}$$

When the model encounters a novel domain (e.g., "pills"), it automatically applies this learned principle: it maps normal and anomalous pills to distinct regions, but crucially, it ensures that the displacement vectors between these regions still adhere to the universal topo-semantic consistency constraint. Because the feature representations for this new domain are organized according to the same structural principle, the simple geometric test $S(k) = \boldsymbol{z}_k \cdot \boldsymbol{w}$ remains effective, leading to clear separation and excellent generalization.

### E.3 Pseudocode for Topological Algorithms

## F Experimental Setup and Implementation Details

### F.1 Detailed Experimental Setup and Implementation

**Datasets and Training Paradigms** Our experiments are conducted across a range of standard anomaly detection benchmarks, encompassing both industrial (MVTec-AD(Bergmann et al., 2021), VisA(Zou et al., 2022), RealIAD(Wang et al., 2024), BTAD(Mishra et al., 2021), MPDD(Jezek et al., 2021)) and medical domains (ColoDB(Tajbakhsh et al., 2016), ClinicDB(Bernal et al., 2015), CVC-300(Bernal et al., 2017), Kvasi(Jha et al., 2019), Retina(Bao et al., 2024), Brain(Bao et al., 2024)). We define three progressive training settings to comprehensively evaluate all methods:

- **S1: Single-Domain Generalization** This setting simulates the most common zero-shot anomaly detection scenario, for which we adopt two tailored protocols based on the evaluation's objective.
    - For the main SOTA comparison across multiple benchmarks (Table 1), we establish a general training protocol. To evaluate all target datasets *except for VisA*, we consistently use the training set of **VisA** as the single source domain, chosen for its complexity and diversity. Conversely, to ensure a fair, non-overlapping evaluation on VisA itself, we exclusively use the **MVTec AD** training set as the source domain.
    - For the performance evolution analysis (Figure 1(d) and Table 2), a consistent testbed is required to fairly compare performance across the S1, S2, and S3 stages. Since our large-scale training data for S2 and S3 contains the MVTec AD dataset, VisA serves as the only uncontaminated testbed for this analysis. Consequently, to maintain a consistent evaluation protocol, the S1 model for this specific analysis is also trained on **MVTec AD** for evaluation on VisA, aligning it with the S2 and S3 models.
- **S2: Multi-Domain, Clean Labels** This setting is designed to explore the model's learning capability on multi-domain data. The training set is composed of approximately 40k images with high-quality annotations from instruct-125k(Xu et al., 2025) (including industrial and medical domains).
- **S3: Large-Scale, Heterogeneous Data** This setting simulates a real-world training scenario with large-scale, noisy data. The training set consists of the 40k clean images from S2, augmented with 85k unlabeled web images from instruct-125k(Xu et al., 2025)

---

**Algorithm 1** Overall TDA-CLIP Framework

---

1: **Part 1: Two-Stage Training**
2: **Input:** Training set $\mathcal{D}_{train}$; Frozen CLIP model $\Phi_{CLIP}$.
3: **Output:** Trained TGA module $\text{TGA}_\phi$; Trained Image Adapter $\text{Adapter}_\theta$.
4: Initialize trainable parameters for TGA ($\phi$) and Adapter ($\theta$).
5:
6: *// Stage 1: Train the Topology-Guided Attention (TGA) Module*
7: **for** each epoch **do**
8:     **for** each batch $X$ in $\mathcal{D}_{train}$ **do**
9:         Update TGA parameters $\phi$ using $\text{UPDATETGA}(X, \Phi_{CLIP}, \phi)$ (see Algorithm 2).
10:     **end for**
11: **end for**
12:
13: *// Stage 2: Train the Image Adapter using Homology Consistency (HC) Loss*
14: **for** each epoch **do**
15:     **for** each batch $(X, L)$ in $\mathcal{D}_{train}$ **do**
16:         Update Adapter parameters $\theta$ using $\text{UPDATEADAPTER}((X, L), \Phi_{CLIP}, \text{TGA}_\phi, \theta)$ (see Algorithm 3).
17:     **end for**
18: **end for**
19:
20: **Part 2: Inference**
21: **Input:** Test sample $x_{test}$; Trained $\text{Adapter}_\theta$; Text prompts.
22: **Output:** Final anomaly map $\boldsymbol{S}_{final}$.
23:                                  ▷ The TGA module is detached/pruned during inference
24:
25: $\boldsymbol{F}_{img}^{test} \leftarrow \text{ForwardPass}_\theta(x_{test})$         ▷ Forward pass using only the trained Adapter
26: $\boldsymbol{t}_{norm}, \boldsymbol{t}_{anom} \leftarrow \text{GetTextFeatures}(\text{"normal"}, \text{"anomaly"})$
27:
28: **for** each patch feature $\boldsymbol{f}_{patch}$ in $\boldsymbol{F}_{img}^{test}$ **do**
29:     $\boldsymbol{S}_{patch} \leftarrow \text{sim}(\boldsymbol{f}_{patch}, \boldsymbol{t}_{anom}) - \text{sim}(\boldsymbol{f}_{patch}, \boldsymbol{t}_{norm})$
30: **end for**
31: Assemble patch scores into the final anomaly map $\boldsymbol{S}_{final}$.
32: **return** $\boldsymbol{S}_{final}$

---

---

**Algorithm 2** TGA Module Update Step

---

1: **function** $\text{UPDATETGA}(X, \Phi_{CLIP}, \phi)$
2:     $\boldsymbol{F}_{inter} \leftarrow \text{ExtractIntermediateFeatures}(\Phi_{CLIP}, X)$
3:     $\boldsymbol{F}_{topo} \leftarrow \text{TGA}_\phi(\boldsymbol{F}_{inter})$              ▷ TGA processes local grid topology
4:
5:     *// Distill topological structure back to the main encoder's path*
6:     $L_{align} \leftarrow \text{ComputeAlignmentLoss}(\boldsymbol{F}_{topo}, \boldsymbol{F}_{inter})$
7:
8:     Update TGA parameters $\phi$ by descending the gradient of $L_{align}$.
9:     **return** updated $\phi$
10: **end function**

---

For the large-scale S2 and S3 settings, our method's hyperparameters were kept identical to those used in the S1 setting without any further tuning. In contrast, a degree of hyperparameter tuning was performed for the baseline methods in an effort to achieve their optimal performance on these larger, more diverse datasets.

**Baselines** We compare our method against several state-of-the-art approaches. To ensure a rigorous and fair comparison, all baseline results are reproduced using their officially released code and pre-trained weights. To maintain a strictly controlled experimental environment, all methods (including our own and all baselines) are built upon the same foundation: the ViT-L/14-336 from

---

**Algorithm 3** HC Loss Calculation and Adapter Update Step

---

1: **function** UPDATEADAPTER($(X, L), \Phi_{CLIP}, \text{TGA}_\phi, \theta$)
2:     $\boldsymbol{F}_{img} \leftarrow$ ForwardPass$_{\theta,\phi}(X, \Phi_{CLIP})$         ▷ Full forward pass with TGA and Adapter
3:     $Z_{txt} \leftarrow$ GetTextFeatures("normal", "anomaly")
4:
5:     *// — Homology Consistency (HC) Loss Calculation —*
6:     $D_{img} \leftarrow$ ComputeDistanceMatrix($\boldsymbol{F}_{img}$)
7:     $\mathcal{P}_{img} \leftarrow$ PersistentHomology($D_{img}$)         ▷ Get persistence pairs
8:     $\mathcal{P}_{clean} \leftarrow$ FilterPairsByLabel($\mathcal{P}_{img}, L$)         ▷ Keep only inter-class pairs
9:
10:     **if** $\mathcal{P}_{clean}$ is not empty **then**
11:         $V_{img}, V_{txt} \leftarrow$ GetVisualAndSemanticTracks($\mathcal{P}_{clean}, \boldsymbol{F}_{img}, Z_{txt}, L$)
12:         $L_{HC} \leftarrow 1 - \text{mean}(\cos(V_{img}, V_{txt}))$
13:     **else**
14:         $L_{HC} \leftarrow 0$
15:     **end if**
16:     Update Adapter parameters $\theta$ by descending the gradient $\nabla_\theta \mathcal{L}_{\text{Stage2}}$ (incorporating $\mathcal{L}_{\text{HC}}$)
17:     **return** updated $\theta$
18: **end function**

---

**Algorithm 4** Computation of Persistence-Critical Pairs $\mathcal{P}_{\text{critical}}$

---

1: **Input:** Feature point cloud $Z = \{\boldsymbol{z}_i\}_{i=1}^{B}$, labels $Y = \{y_i\}_{i=1}^{B}$.
2: **Output:** Set of persistence-critical pairs $\mathcal{P}_{\text{critical}}$.
3:
4: Initialize edge list $E \leftarrow []$.
5:                            ▷ 1. Construct graph edges with their weights (distances)
6: **for** $i = 1$ to $B$ **do**
7:     **for** $j = i + 1$ to $B$ **do**
8:         $d_{ij} \leftarrow \|\boldsymbol{z}_i - \boldsymbol{z}_j\|_2$.
9:         Append $(i, j, d_{ij})$ to $E$.
10:     **end for**
11: **end for**
12:
13: Sort $E$ in ascending order based on distances $d_{ij}$.
14:
15:                  ▷ 2. Find 0-dim death edges using a Union-Find data structure
16: Initialize Union-Find structure 'uf' with $B$ disjoint sets, one for each point.
17: Initialize death edge indices $E_{\text{death\_indices}} \leftarrow []$.
18: **for** each edge $(i, j, d_{ij})$ in sorted $E$ **do**
19:     **if** 'uf.find(i)' $\neq$ 'uf.find(j)' **then**
20:         Append pair $(i, j)$ to $E_{\text{death\_indices}}$.
21:         'uf.union(i, j)'.
22:     **end if**
23: **end for**
24:
25:                                 ▷ 3. Filter for cross-class pairs
26: Initialize $\mathcal{P}_{\text{critical}} \leftarrow []$.
27: **for** each pair $(i, j)$ in $E_{\text{death\_indices}}$ **do**
28:     **if** $y_i \neq y_j$ **then**
29:         Append pair $(i, j)$ to $\mathcal{P}_{\text{critical}}$.
30:     **end if**
31: **end for**
32: **return** $\mathcal{P}_{\text{critical}}$.

---

CLIP is used as the backbone, and all input images are uniformly resized to 518x518 during both training and inference.

**Evaluation Metrics**   We use standard metrics in the field of anomaly detection. For image-level classification, we report the Area Under the Receiver Operating Characteristic curve (Image-AUROC). For pixel-level localization, we report both Pixel-AUROC and the Area Under the Precision-Recall curve (Pixel-AUPR), the latter being particularly informative for datasets with imbalanced anomaly sizes.

**Training Environment**   We use the AdamW optimizer with an initial learning rate of $1 \times 10^{-4}$ and a cosine annealing schedule over the course of training. The batch size was set to 64 for all experiments.

**Model Architecture**   Our model integrates lightweight Adapter modules into the CLIP backbone. Specifically, adapters for the text encoder are inserted into the first 3 Transformer layers, while adapters for the visual encoder are inserted into the first 4 Transformer layers.

**Hyperparameter Settings**   Key hyperparameters for our method are set as follows unless specified otherwise in sensitivity analyses. For the TGA module, the local grid size is $k = 4$ and the softmax temperature is $\tau = 1.0$. The knowledge distillation fusion weight is set to $\alpha = 0.1$. For the loss functions, the alignment loss weight is $\lambda_{\text{align}} = 0.1$, and the homology consistency loss weight is $\lambda_{\text{hc}} = 0.1$, seg loss weight is $\lambda_{\text{seg}} = 1$, cls loss weight is $\lambda_{\text{cls}} = 0.5$.

### F.2   Hyperparameter Settings and Analysis for Baseline Methods

To ensure our experimental comparisons are fair and rigorous, especially under the novel large-scale S2 (multi-domain, clean) and S3 (large-scale, heterogeneous) settings, we conducted dedicated adjustments and hyperparameter searches for the two primary baseline methods: AnomalyCLIP and AA-CLIP. Our goal was to identify the optimal configuration for each method on these more challenging datasets to guarantee that their reported performance degradation was not an artifact of suboptimal parameter settings.

#### F.2.1   Adjustments to the Training Strategy for the S3 Dataset

Before conducting the specific hyperparameter search, we first addressed a key challenge posed by the S3 dataset. The S3 setting augments the S2 data with an additional 85,000 web images, including both normal and anomalous samples, that lack pixel-level mask annotations. This presented a problem for all baseline methods that rely on a segmentation loss: if a training batch consisted entirely of images without masks, the segmentation loss term for that batch would become invalid, leading to unstable gradients and training failure.

To create a fair and effective training environment for all baseline methods, we uniformly adopted the following strategy for them: First, we increased the batch size and designed a batch sampling strategy to enforce that every batch contains *at least one image with a mask annotation*. Second, during the loss calculation within a batch, our implementation automatically skips the samples that lack masks, computing the loss only on the annotated samples. These measures ensured that the gradient from the segmentation loss remained consistently valid, allowing the baseline models to undergo stable and meaningful training on the semi-supervised S3 dataset.

#### F.2.2   AnomalyCLIP

For the S1 (single-domain) setting, we strictly followed the optimal hyperparameters reported in the original AnomalyCLIP paper. Acknowledging that the S2 and S3 training data are significantly more complex, we inferred that the original number of training epochs and the temperature coefficient of the segmentation loss might not be optimal. We therefore focused on a grid search over these two key parameters. The search space for the number of epochs was set to 1, 2, 3, 4, 5, 10, 15, and the search space for the segmentation loss temperature was 0.01, 0.05, 0.07,0.1,0.2,0.5,1.

We found that for the S3 setting, increasing the number of epochs to 5 and adjusting the temperature to 0.5 yielded the best performance on our validation set. Consequently, the results reported in Table 2 are based on these optimized hyperparameters. Despite this tuning, the performance of AnomalyCLIP still dropped noticeably in the S3 setting, suggesting that its learned single object-agnostic prompt struggles to generalize across heterogeneous domains.

### F.2.3 AA-CLIP

Similar to AnomalyCLIP, the S1 setting for AA-CLIP used the default hyperparameters from its official implementation. For the S2 and S3 settings, we conducted a detailed search over the number of training epochs and the segmentation loss temperature. The search space for text epochs was set to 1, 2, 5, and image epochs was set to 5,10,15,20, and for the temperature coefficient, it was 0.01, 0.05, 0.07,0.1, 0.2, 0.5, 1.

Through evaluation, we identified the combination of 2 training text epochs,10 image epochs, and a temperature of 0.07 as the best-performing configuration. The results we report in Table 2 are based on this setup. It is noteworthy that even with this specialized training strategy and parameter optimization, the performance of AA-CLIP still suffered a catastrophic decline in the S3 setting. This strongly supports our core hypothesis: when faced with multi-domain data having complex and conflicting topological structures, the performance bottleneck of methods relying on a single, rigid geometric objective stems from their intrinsic methodological limitations.

In summary, through the dedicated adjustments to the training strategy and the detailed hyperparameter tuning process described above, we are confident that the performance decline of the baseline methods in the S2 and S3 settings is a genuine reflection of their inherent limitations in handling large-scale heterogeneous data, which we define as the Domain Conflict, and is not a result of an unfair experimental setup.

## G ADDITIONAL EXPERIMENTAL ANALYSES

### G.1 HYPERPARAMETER SENSITIVITY

To validate the robustness of our framework, we conducted sensitivity analyses for our key hyperparameters. All experiments were performed on the MVTec AD dataset under the S1 setting. Table 5 shows the impact of the loss weights for $\mathcal{L}_{HC}$ ($\lambda_{hc}$) and $\mathcal{L}_{align}$ ($\lambda_{align}$). Tables 6 and 7 show the sensitivity for the TGA module's local grid size $k$ and softmax temperature $\tau$, respectively. The results demonstrate that our model's performance is stable across a reasonable range of values for all tested hyperparameters, indicating low sensitivity to their precise settings.

Table 5: Hyperparameter sensitivity analysis on the MVTec AD dataset. We evaluate the impact of the loss weights for $\mathcal{L}_{HC}$ ($\lambda_{hc}$) and $\mathcal{L}_{align}$ ($\lambda_{align}$). The model shows stable performance across a reasonable range of values for both hyperparameters.

| Hyperparameter | Value | Pixel-AUPR | Pixel-AUROC | Image-AUROC |
|---|---|---|---|---|
| $\lambda_{hc}$ for $\mathcal{L}_{HC}$ | 0 (Baseline) | 45.6 | 92.1 | 91.8 |
| | 0.1 | **48.3** | **92.2** | 91.9 |
| | 0.2 | 48.0 | 92.1 | **92.0** |
| $\lambda_{align}$ for TGA | 0 | 48.3 | **92.2** | 91.9 |
| | 0.1 | **48.6** | **92.2** | **93.2** |
| | 0.2 | 47.2 | 91.5 | 92.3 |
| | 0.5 | 47.4 | 91.4 | 92.8 |

Table 6: Sensitivity analysis for the TGA grid size $k$ on the MVTec AD dataset. The default value used in our main experiments is highlighted in **bold**.

| Hyperparameter | Value | Pixel-AUPR | Pixel-AUROC | Image-AUROC |
|---|---|---|---|---|
| TGA Grid Size ($k$) | **4** | **48.3** | **92.2** | **93.2** |
| | 6 | 47.8 | 91.6 | 92.5 |
| | 8 | 47.5 | 91.5 | 92.3 |

Table 7: Sensitivity analysis for the TGA softmax temperature $\tau$ on the MVTec AD dataset. The default value used in our main experiments is highlighted in **bold**.

| Hyperparameter | Value | Pixel-AUPR | Pixel-AUROC | Image-AUROC |
|---|---|---|---|---|
| | 0.5 | 47.6 | 92 | 92.1 |
| TGA Temperature ($\tau$) | **1.0** | **48.3** | **92.2** | **92.3** |
| | 2.0 | 46.9 | 91.8 | 91.9 |

### G.2 GENERALIZABILITY AS A PLUG-AND-PLAY MODULE

To conclusively demonstrate that our proposed components are general-purpose, plug-and-play tools rather than being tailored only to our specific baseline, we integrated our topological tools into a strong state-of-the-art method, AA-CLIP. We followed the original training procedure of AA-CLIP, merely augmenting its final training objective with our $\mathcal{L}_{\text{HC}}$ and/or TGA module.

The results, presented in Table 8, are unequivocal. On the challenging VisA dataset, the simple addition of the lightweight $\mathcal{L}_{\text{HC}}$ brought significant performance gains to AA-CLIP (e.g., a 0.3% improvement in Image-AUROC). When both TGA and $\mathcal{L}_{\text{HC}}$ were added, its Pixel-AUPR increased substantially from 45.3% to **47.8%** (**+2.5%**), fully demonstrating that our macro- and micro-level tools can work effectively in synergy on other architectures. This experiment strongly validates our core thesis: topological alignment is a versatile and effective regularization strategy, and our proposed tools can serve as a ready-to-use toolkit for enhancing the robustness of a wide range of vision-language models in the ZS-AD domain.

Table 8: Validation of the generalizability of our topological tools. Applying them to a strong SOTA baseline (AA-CLIP) on the VisA dataset yields significant improvements, demonstrating their utility as plug-and-play modules.

| Method on MVTecAD Dataset | Pixel-AUPR | Pixel-AUROC | Image-AUROC |
|---|---|---|---|
| AA-CLIP (Original) | 45.3 | 91.9 | 90.5 |
| AA-CLIP + $\mathcal{L}_{\text{HC}}$ | **45.8** (+0.5) | 91.9 (+0.0) | **90.8** (+0.3) |
| AA-CLIP + TGA | 47.0 (+1.7) | 91.5 (-0.4) | **91.5** (+1.0) |
| AA-CLIP + Both | **47.8** (+2.5) | **92.1** (+0.2) | 91.3 (+0.8) |

### G.3 ANALYSIS ON THE IMPACT OF HIGHER-DIMENSIONAL HOMOLOGY

Our main framework relies on 0-dimensional homology ($H_0$), which captures the connected components of the feature space. A natural question is whether higher-dimensional topological features, such as the 1-dimensional loops ($H_1$), could bring additional benefits. To examine this, we extended our Homology Consistency Loss ($\mathcal{L}_{HC}$) to jointly constrain both $H_0$ and $H_1$, and compared this variant against our default design (TDA-CLIP using $H_0$ only). Results across industrial and medical benchmarks are summarized in Table 9.

Table 9: Comparison of average performance for different model configurations

| Average Scope | Method | Image-AUROC (%) | Pixel-AUROC (%) | Pixel-AUPR (%) |
|---|---|---|---|---|
| Industrial Mean | TDA-CLIP ($H_0$ only) | 84.6 | 95.4 | 37.2 |
| | TDA-CLIP ($H_0 + H_1$) | 83.8 | 95.2 | 36.3 |
| Medical Mean | TDA-CLIP ($H_0$ only) | 83.4 | 93.3 | 54.0 |
| | TDA-CLIP ($H_0 + H_1$) | 80.4 | 93.2 | 53.2 |
| Overall Mean | TDA-CLIP ($H_0$ only) | 84.2 | 93.7 | 46.3 |
| | TDA-CLIP ($H_0 + H_1$) | 82.8 | 93.6 | 45.4 |

From Table 9, the conclusion is clear: incorporating $H_1$ does not yield consistent gains. On the *industrial datasets*, performance with $H_0 + H_1$ actually degrades across all three metrics (e.g.,

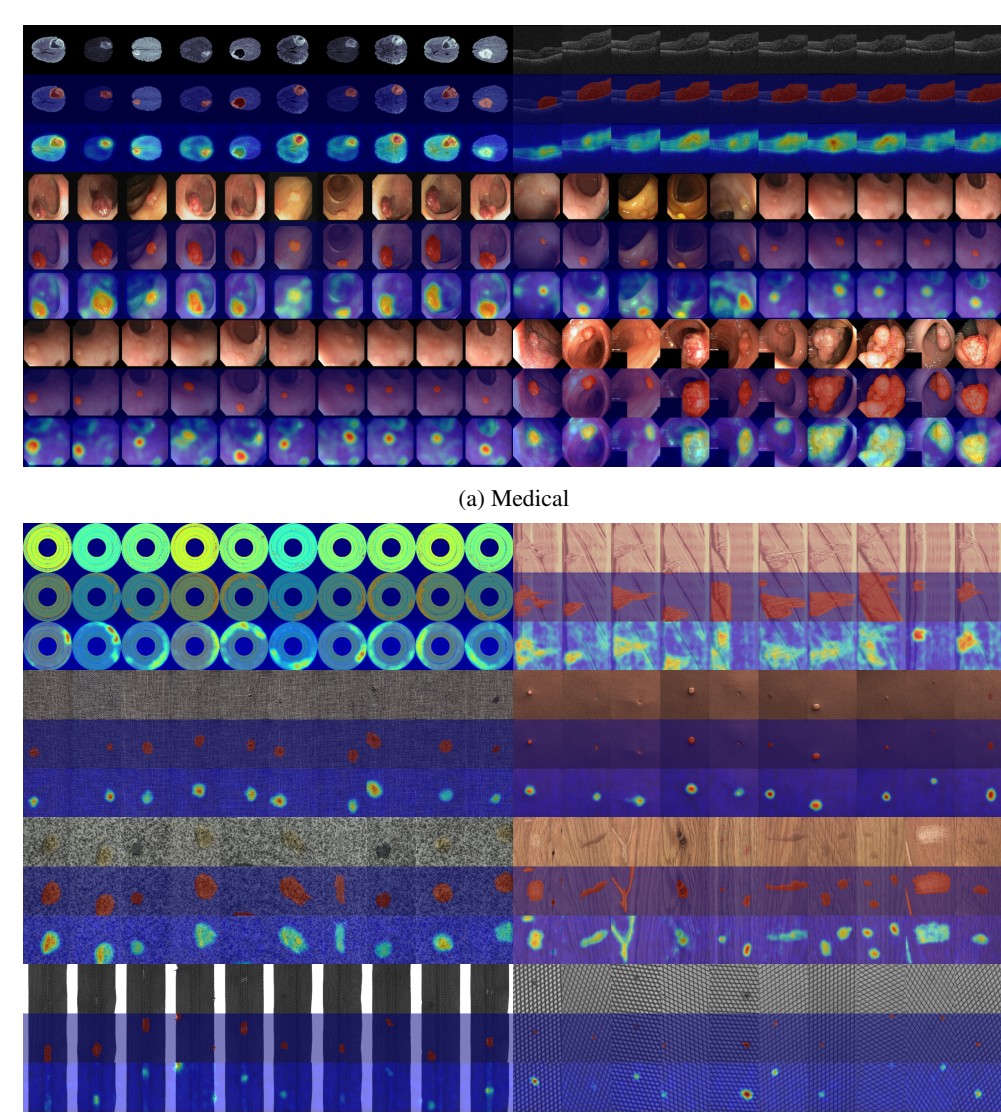

(a) Medical

(b) Texture

Figure 4: Visualizations of anomaly detection results.

Pixel-AUPR drops from 37.2% to 36.3%). On the *medical datasets*, $H_0 + H_1$ achieves nearly the same pixel-level scores as $H_0$ alone, but without improvements at the image level (Image-AUROC decreases by 3.0 points). The *overall averages* follow the same trend, confirming that the potential benefit of $H_1$ is not statistically robust.

Given that computing $H_1$ features introduces substantial additional overhead, while providing no reliable performance improvement (and sometimes harming results), our final design focuses exclusively on the efficient and consistently effective $H_0$ constraint.

# H    ADDITIONAL VISUALIZATIONS

## H.1    VISUALIZATIONS OF ANOMALY DETECTION

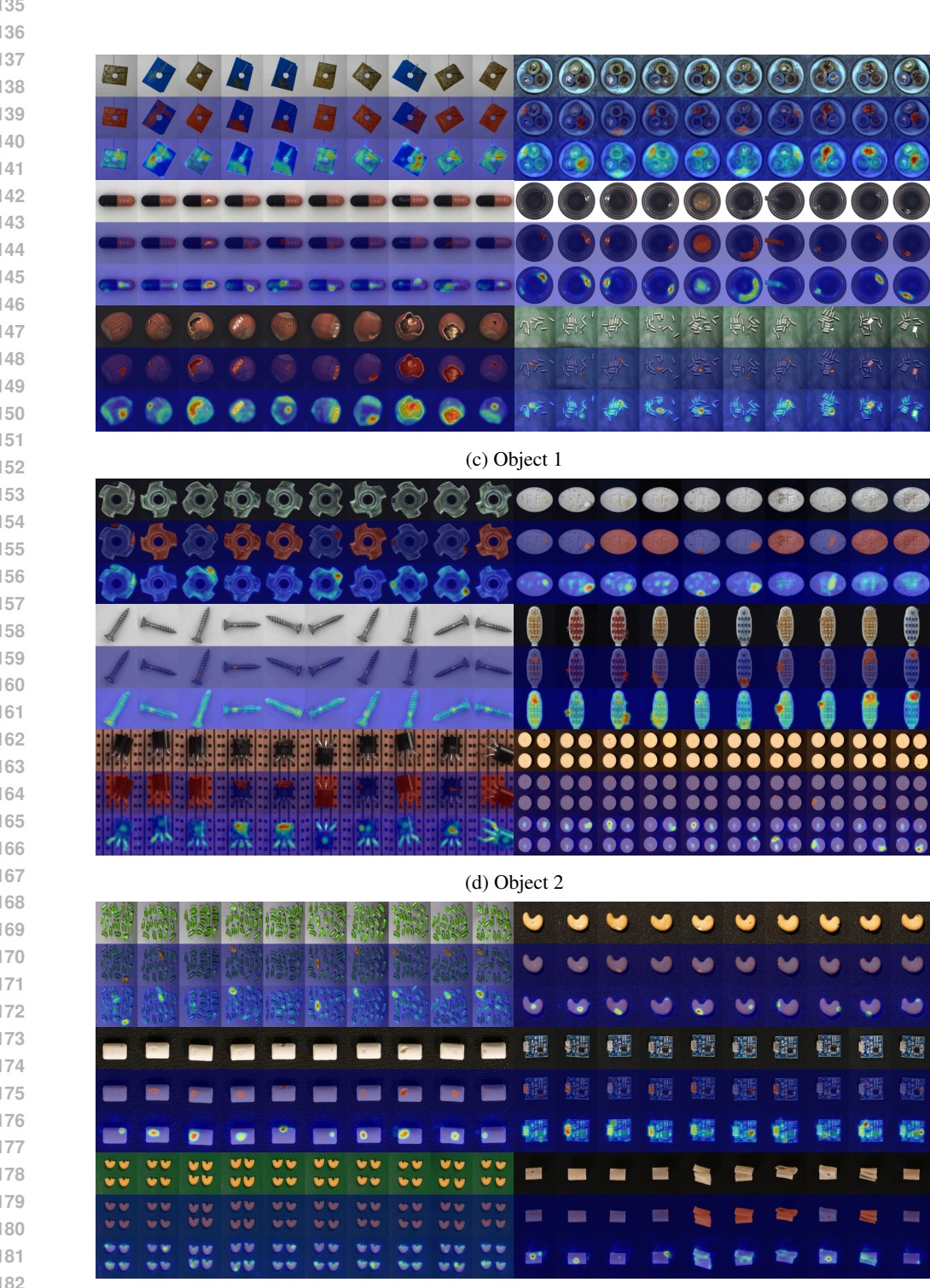

(c) Object 1

(d) Object 2

(e) Object 3

Figure 4: Visualizations of anomaly detection results (continued).

## H.2 STRATEGIES COMPARISON VISUALIZATIONS

To provide further insight into our proposed framework, this section presents two sets of supplemental visualizations. First, Figure 5 offers a direct visual comparison of how different pair mining strategies—including our persistence-critical method and various baselines—operate within the original CLIP feature space for a representative class.

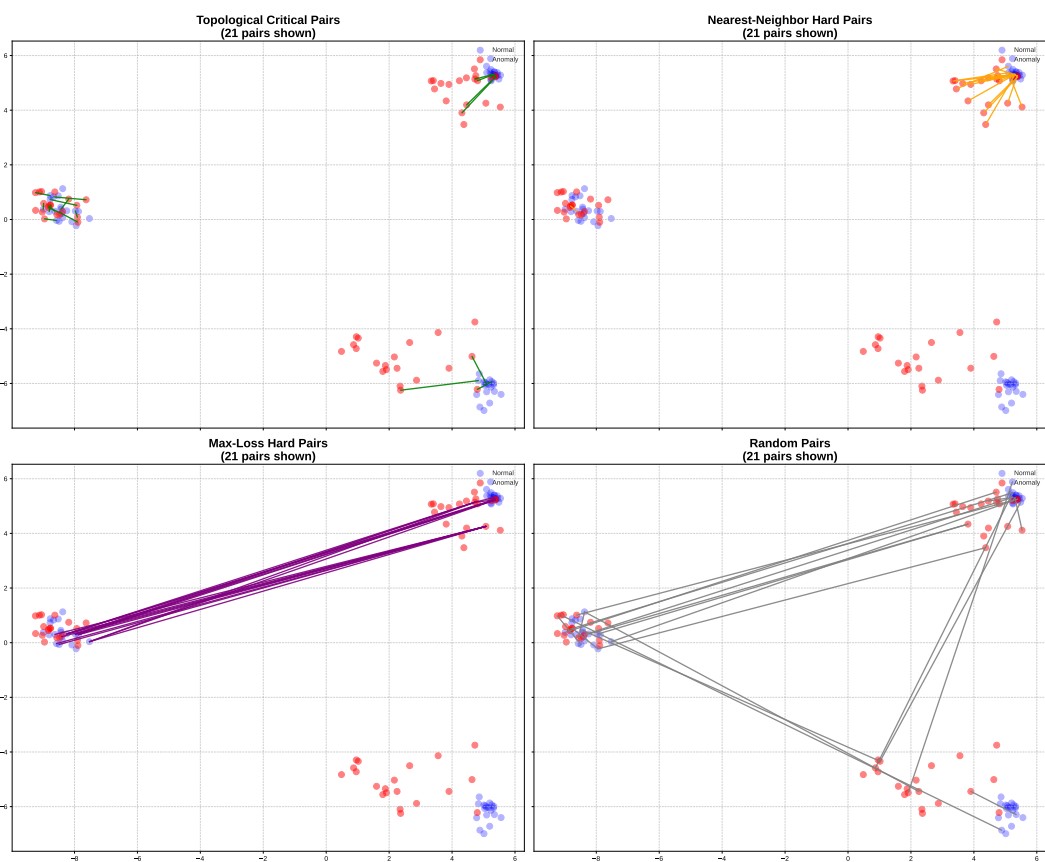

Figure 5: **Visual comparison of different pair mining strategies in the original CLIP feature space**. Each subplot shows the same set of normal (blue) and anomaly (red) image features. The lines represent the pairs selected for training by each strategy. **(a) Persistence-Critical Pairs (Ours):** Selects structurally important "bridge" pairs that connect the normal and anomalous manifolds, providing a clear and robust learning signal. **(b) Nearest-Neighbor Hard Pairs:** Selects geometrically closest pairs, which are often at the dense edges of the manifolds and may not be the most informative for global separation. **(c) Max-Loss Hard Pairs:** Selects pairs that are maximally distant, often involving outliers, which can lead to a brittle decision boundary. **(d) Random Pairs:** Serves as a baseline, showing no structural preference. Our topological method clearly identifies the most informative pairs for separating the two classes.

