# OpenReview forum: "Topological Alignment: A Universal Framework for Anomaly Detection"
_ICLR.cc/2026/Conference — ICLR 2026 Conference Withdrawn Submission_

### Official Review · Reviewer_296f · 2025-10-26

**Soundness:** 2
**Presentation:** 3
**Contribution:** 2
**Rating:** 2
**Confidence:** 5

**Summary:**

This paper addresses the issue of ‘Negative Transfer in Domain Generalization’ for zero-shot anomaly detection based on vision-language models like CLIP. The authors attribute performance degradation under large-scale, diverse-domain training to what they term a ‘Domain Conflict’, and propose TDA-CLIP, a topology-inspired training framework. Two main topological tools are introduced: Homology Consistency Loss (LHC) and Topology-Guided Attention (TGA). Experiments on 11 industrial and medical datasets show moderate improvements in zero-shot anomaly detection performance, maintaining the robustness when training data scales up, while adding no inference overhead.

**Strengths:**

1.	The paper is well structured, includes detailed appendices and reproducibility statements.
2.	The authors evaluate across multiple datasets and domain scales, achieving SOTA metrics among current CLIP-based zero-shot anomaly detection approaches.

**Weaknesses:**

1.	The motivation of this work is somewhat puzzling. For real-world application of anomaly detection, it is incredible to fuse industrial data and medical data to train a universal model. The domain conflicts between these different types of data obvious. Mixing them together significantly increases ‘Domain Conflict’ and the difficulty of model training. In my opinion, there is no need to do so for a specialized model used in practical scenarios.
2.	Although the paper claims a paradigm shift from geometry to topology, the actual method essentially adds a pairwise cosine alignment regularizer based on selected sample pairs. The use of persistent homology to identify these pairs is incremental, and the proposed losses are not fundamentally new compared to existing contrastive or consistency-based objectives.
3.	The reported improvements (typically <1% in AUROC or AUPR) are small and within the range of variance for large-scale vision models. Given the significant additional complexity during training, the performance gains do not convincingly demonstrate that topology-based regularization is the main cause of improvement.
4.	The ‘Persistence-Critical Pairs’ appears to be incremental compared with other pair mining strategies. Given that the total training time is about 20% longer as stated, but the Pixel-AUROC is increase by only 0.1% as shown in Table 3.
5.	The manifold evolution is difficult to understand only with Fig.1. It is hard to see that the proposed regularizer actually modifies topology as claimed. The methodology remains abstract and unverifiable from the results.
6.	The authors declared that their TDA-CLIP consistently benefits from more data in Table 2. It is not very convincing because the improvements of some metrics across different datasets did not reflect consistency. Furthermore, no statistic evaluation was performed to demonstrate this statement.

**Questions:**

Except for the major concerns presented above in the Weaknesses, I have another two additional questions:

1.	Fig. 1 is difficult to understand. What’s the meaning of the arrows in Fig. 1? Compared with Figure 1(a) and (b), what is the finding of the proposed method?

2.	In Fig. 1(d), the performance of image-level AUROC is inconsistent with that of the other two metrics with the increase of training data, what is the reason? Can the robustness be purely attributed to the proposed TDA-CLIP, not affected by the training data itself?

---

### Official Review · Reviewer_KLbQ · 2025-10-30

**Soundness:** 3
**Presentation:** 3
**Contribution:** 2
**Rating:** 2
**Confidence:** 4

**Summary:**

This paper tackles an important and recognized problem in ZS-AD: the severe performance collapse (termed Negative Transfer in Domain Generalization), where SOTA models like AnomalyCLIP and AA-CLIP suffer when trained on large-scale, diverse data.

**Strengths:**

1. The paper's main contribution claim is that TDA-CLIP is the first to overcome NTDG, showing robust performance gains in the S3 (large-scale, heterogeneous) setting where all baselines catastrophically fail.
2. The authors diagnose the root cause of this failure as "Domain Conflict", a fundamental inability of a single, rigid geometric decision boundary to reconcile the complex topological manifolds from multiple, heterogeneous domains.

**Weaknesses:**

1. The primary claim is based on a confounded and unequal comparison. The S3 setting is built by augmenting the S2 (40k annotated images) dataset with 85k "unlabeled web images". The paper admits in Appendix F.2.1 that all baseline methods rely on a "segmentation loss". For the 85k unlabeled images, this loss is invalid. The authors' strategy for the baselines was to "automatically skips the samples that lack masks, computing the loss only on the annotated samples". This means TDA-CLIP is designed as a semi-supervised method that can effectively leverage the entire 125k S3 dataset (using image-level labels and unsupervised topological structures). The baselines, however, are restricted to being supervised methods that are effectively still only training on the 40k S2 subset. Therefore, the performance gap observed in S3 (Table 2) does not prove that TDA-CLIP is more robust to "Domain Conflict." It most likely proves that a semi-supervised method (TDA-CLIP) benefits more from an additional 85k of (partially) unlabeled data than a supervised method (the baselines) that cannot use that data at all.
2. TGA is based on the "Topological Outlier Hypothesis": anomalous patch features should be "topologically isolated" from the "mainland" of normal patches. This is visually demonstrated in Figure 3 with a local defect (a single red-starred patch) on a screw . However, this hypothesis may completely fail for global or textural anomalies (e.g., "carpet," "grid," or "leather" classes in MVTec AD). In those cases, all patches may belong to the "anomalous" class, forming their own dense manifold, not "topological islands." TGA's saliency calculation, which relies on isolation, would likely fail or provide a misleading signal for these common anomaly types.
3. The computational cost of TGA is significantly downplayed. The module operates per-image19. It partitions the patch grid into $k^2$ disjoint subsets (where $k=4$) and then computes 0-dimensional persistent homology in parallel for each local point cloud. This means separate persistent homology computations (Algorithm 4) are required per-image, per-batch, during training. The claim that this adds only a "modest overhead" and that "total training time is only about 20% longer" seems highly optimistic for such a computationally intensive operation and lacks a rigorous FLOPs analysis.
4. The paper claims this complex design "avoids conflicting gradients from simultaneous optimization". This is a generic claim. What gradients, specifically, conflict? Why would a simpler, end-to-end training with a standard (perhaps weighted) multi-task loss fail?
5. The Stage 2 design is particularly confusing. The TGA ("teacher") processes patch features from the visual adapter ("student"), and its output $z_{tda}$ is then added back to the "student's" own global average pooled feature $z_{avg}$ to create a "fused feature" $z_{out}$. This fused feature is then used for the auxiliary losses ($L_{align}, L_{HC}$). This "student-teacher-student" feedback loop is non-standard and its motivation is unclear.
6. The ablation in Table 4 only compares Baseline, +$L_{HC}$, +TGA, and +Both. It does not ablate the necessity of the two-stage decoupled strategy itself. A crucial missing experiment is a single-stage, end-to-end trained model with all components. It also fails to ablate the complex $z_{out}$ fusion mechanism in Stage 2.

**Questions:**

See weaknesses.

---

### Official Review · Reviewer_3JEC · 2025-11-01

**Soundness:** 2
**Presentation:** 2
**Contribution:** 2
**Rating:** 2
**Confidence:** 4

**Summary:**

This work tackles the challenging task of zero-shot anomaly detection. It identifies a source of negative transfer in domain generalization, arising from a fundamental domain conflict caused by incompatible manifold structures between normal and anomalous data. To address this issue, it introduces TDA-CLIP, which combines a homology-based global loss with topology-guided attention to enforce topological separability and improve anomaly detection performance.

**Strengths:**

The empirical results are strong, indicating that incorporating topological biases can effectively mitigate negative transfer in anomaly detection.

Additionally, the two-stage, adapter-based training strategy is efficient.

**Weaknesses:**

The “Domain Conflict” explanation is insufficiently proven; evidence for topology as the root cause is limited and mostly conceptual.

The claim regarding large-scale cross-domain training is not convincingly supported by the presented experiments.

The title’s use of “Universal” overstates the scope; universality is not rigorously demonstrated and training domains remain limited.

In Line 96, “current methods” is vague—specific baselines and approaches should be explicitly referenced.

The paper does not cite foundational persistent homology literature, which is necessary given the central role of topological tools.

“Traditional geometric hard-mining” is not clearly defined, and clearer comparison and intuition are needed to understand the improvement brought by persistent homology.

Figures (e.g., Figure 3) contain text that is too small to read, which hinders clarity, and more experimental detail is required for reproducibility.

**Questions:**

How sensitive is performance to hyperparameters?

It is unclear why the adapter is inserted specifically at the first layer of the encoders.

---

### Official Review · Reviewer_6Pwd · 2025-11-04

**Soundness:** 3
**Presentation:** 2
**Contribution:** 3
**Rating:** 4
**Confidence:** 3

**Summary:**

This paper considers CLIP-based anomaly detection. The proposed approach attempts to preserve topological structure of CLIP embedding space rather than enforcing strict geometrical separability between normal and anomalous samples during finetuning, which becomes challenging once applied to mulitple domains. This is done by using by first trains the model to align the direction of vectors connecting normal and anomalous sample image embeddings with vectors that connect textual embeddings of normal and anomalous sample descriptions. A further feature space refinement is achieved by adding loss compontents that indentify most salient image patches that are most informative for anomaly detection.

**Strengths:**

[S1] The proposed method achieves good experimental results on diverse set of anomaly detection approaches, beating previous CLIP based methods.

[S2] The idea of not enforcing geometric separability, but more flexible constraints is interesting.

**Weaknesses:**

[W1] I do not like the framing of the proposed approach as a universal framework for anomaly detection. This approach is specific to CLIP based methods, and is therefore limited by the capabilities of the CLIP model (e.g. is it possible to detect real anomalies with respect to CLIP training data) Furthermore, there is no comparison to methods that do not rely on CLIP, but the results could still be lagging. E.g. for MVTec the results are worse with the CLIP base approaches [a].

[W2] I found the paper to be difficult to read and some of the claims are not particularly convincingly substantiated. See questions.

[W3] The paper relies on existence on anomalous samples during training, which is not necessarily needed in considered area of anomaly detection.

[a]  Mousakhan el al., Anomaly detection with conditioned denoising diffusion models, GCPR 2024

**Questions:**

[Q1] Figure 1 claims that it demonstrates that normal and anomalous samples are consistently separated in terms of direction. does that refer to individual datasets in the collection. Overall the colors are not clear, it is not obvious what the clusters are supposed to represent.

[Q2] The paper claims that the proposed approach benefits from more data, but that is not completely true. as demonstrated by Figure 1 (e) (AP performance slighlty drops for on the largest dataset collection.

[Q3] To my understanding, the anomaly detection results for larger dataset collections are calculated as means over individual dataset collections. What would happen if those datasets were really merged?

[Q4] HOw were the textual embeding for normal and anomalous samples designed? How sensitive it the emthod to this design?

[Q5] The method relies on balancing between different losses. How were these values chosen?

[Q6] What is loss seg? What is the ground truth for it?

---

### Note · Authors · 2025-11-12

I have read and agree with the venue's withdrawal policy on behalf of myself and my co-authors.